# UniTox: Leveraging LLMs to Curate a Unified Dataset of Drug-Induced Toxicity from FDA Labels

**Jake Silberg**
Stanford University
`jsilberg@stanford.edu`

**Kyle Swanson**
Stanford University
`swansonk@stanford.edu`

**Elana Simon**
Stanford University
`epsimon@stanford.edu`

**Angela Zhang**
Stanford University
`angelaz@stanford.edu`

**Zaniar Ghazizadeh**
Stanford University
`zaniar@stanford.edu`

**Scott Ogden**
Genmab
`scog@genmab.com`

**Hisham Hamadeh**
Genmab
`hha@genmab.com`

**James Zou**
Stanford University
`jamesz@stanford.edu`

## Abstract

Drug-induced toxicity is one of the leading reasons new drugs fail clinical trials. Machine learning models that predict drug toxicity from molecular structure could help researchers prioritize less toxic drug candidates. However, current toxicity datasets are typically small and limited to a single organ system (e.g., cardio, renal, or liver). Creating these datasets often involved time-intensive expert curation by parsing drug labelling documents that can exceed 100 pages per drug. Here, we introduce UniTox[1], a unified dataset of 2,418 FDA-approved drugs with drug-induced toxicity summaries and ratings created by using GPT-4o to process FDA drug labels. UniTox spans eight types of toxicity: cardiotoxicity, liver toxicity, renal toxicity, pulmonary toxicity, hematological toxicity, dermatological toxicity, ototoxicity, and infertility. This is, to the best of our knowledge, the largest such systematic human *in vivo* database by number of drugs and toxicities, and the first covering nearly all non-combination FDA-approved medications for several of these toxicities. We recruited clinicians to validate a random sample of our GPT-4o annotated toxicities, and UniTox's toxicity ratings concord with clinician labelers 85–96% of the time. Finally, we benchmark several machine learning models trained on UniTox to demonstrate the utility of this dataset for building molecular toxicity prediction models.

## 1 Introduction

An estimated 90% of drugs fail in clinical trials [1]. While the most common cause of failure is efficacy, one study found that the second largest cause (24% of failures) was drug safety [2]. Further, every year, previously approved drugs are taken off the market as unanticipated toxicities become apparent in post-marketing data that can be difficult to screen pre-clinically [3]. These different drug-induced toxicities span many different organ systems, including the heart, liver, kidneys, blood, and lungs. As a result, there is a strong need for predictive models that can anticipate a broad range

---

[1]UniTox data is available at `https://zou-group.github.io/UniTox-website`. Code available at: `https://github.com/jsilbergDS/UniTox`

38th Conference on Neural Information Processing Systems (NeurIPS 2024) Track on Datasets and Benchmarks.

of human *in vivo* toxicities so that researchers can screen for molecules with the highest chance of clinical trial and post-market safety and success.

A major source of both data and expertise in evaluating drug-induced toxicity is the FDA. One critical function of the FDA is to approve drug labels, which we define here to avoid confusion with "label" in the machine learning context. An FDA Drug Label is a comprehensive regulatory document written in collaboration between the FDA and the pharmaceutical company seeking drug approval. It contains all the information that clinical prescribers and/or patients taking this medicine might want to know, such as Indications and Dosages (what the drug should be taken to treat), Warnings and Precautions (any suspected risks of taking the drug), and a summary of the efficacy and safety results from all Clinical Trials reviewed by the FDA as part of their decision to approve the drug. FDA Drug Labels are typically about 10-20 pages long, but some FDA Drug Label documents can be over 100 pages long. The FDA continuously revises these drugs labels as ongoing benefits and safety risk information about a drug becomes available after the initial drug approval.

FDA researchers have published analyses of drug labels on drug-induced cardiotoxicity (DICTrank [4]), drug-induced liver injury (DILIrank [5]), and drug-induced renal toxicity (DIRIL [6]). Each analysis has involved one or more trained professionals who carefully comb through each drug label to make a toxicity determination.

More recently, the FDA has explored the use of large language models (LLMs) to process drug labels more quickly [7]. They developed askFDALabel, a retrieval-augmented generation (RAG) [8] system that finds the most similar drug label fragments to a user query, then utilizes a fine-tuned LLM to generate a response based on those fragments. They showcase askFDALabel for assessing drug-induced cardiotoxicity (DICT) and find that, where ratings were available, askFDALabel agrees with the human-annotated dataset 78% of the time.

In addition to that work, several other toxicity databases have been developed. For example, Cavasotto and Scardino [9] compiled a set of toxicity databases. These existing datasets have several limitations. First, these datasets are often small due to time-consuming annotation efforts [6]. Second, these datasets use different methodologies to evaluate toxicities. For example, the FDA's DIRIL (renal toxicity) work draws on two existing datasets that disagreed more than 30% of the time on the same drugs [10, 11]. Many of these, such as SIDER [12], ECHA's C&L system [13], PubChem's Hazardous Substances Data Bank [14], and the Comparative Toxicogenomics Database [15], cannot be used to search by toxicity status and do not include all toxicity keywords in their side effects or phenotype data. While Tox21 and ToxCast [16] cover a large number of chemicals, not limited to FDA-approved medications, they are based on *in vitro* assays that may not accurately reflect *in vivo* drug effects. These chemical databases also typically exclude biologics. Other very comprehensive toxicity databases, such as PNEUMOTOX [17] for pulmonary toxicity and LiverTox [18] for liver toxicity, cover only a single organ system and may differ in methodologies. Machine learning models for toxicity that are trained on these datasets [19, 20, 21, 22, 23], while useful, suffer from the same limitations as the underlying datasets.

**Our contributions.** In this work, we develop a framework for using LLMs to rapidly categorize the toxicity of drugs from FDA drug labels. We apply this methodology to build UniTox, the largest human *in vivo* cross-toxicity dataset of 2,418 FDA-approved drugs. We evaluate the accuracy of these predictions, achieving up to 93% accuracy on pre-existing datasets compared to 78% for askFDALabel, and as well as up to 85–96% concordance on a clinician-reviewed sample. Finally, we benchmark the performance of several machine learning models trained on small molecule drugs from UniTox to illustrate the benefit of building a uniform toxicity dataset.

## 2 Methods

### 2.1 Building UniTox

To build UniTox, we first needed to curate a set of drugs and associated drug labels to analyze. Drawing inspiration from askFDALabel, we started with the universe of all non-combination human prescription drugs from the FDALabel database [24]. One important difference from askFDALabel is that we included biologic drugs, as those were included in DICTrank. We then grouped drugs by unique generic drug names and removed labels where the route of administration included topical, irrigational, or intradermal. For each unique generic drug name where we had an exact match with

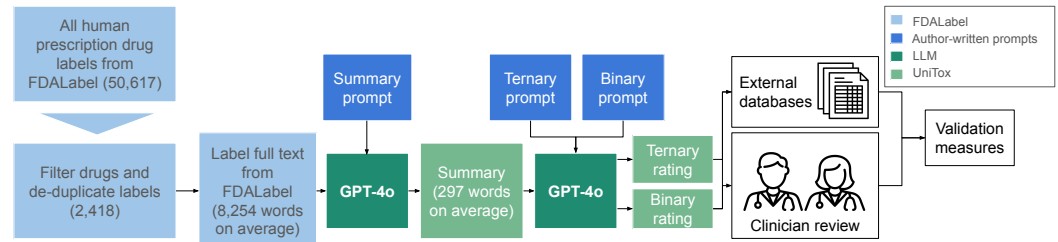

Figure 1: UniTox was built by applying a large language model (GPT-4o) to a curated set of 2,418 FDA drug labels to produce ternary (No/Less/Most) and binary (No/Yes) toxicity ratings, which were evaluated based on external databases and clinican review.

askFDALabel, we used the same drug label. Where we did not have an exact match, we used the most recent New Drug Application (NDA) drug label for that generic drug name. Where we did not have an NDA drug label, we used the most recent Abbreviated New Drug Application (ANDA) drug label, which is used for generic versions of brand-name drugs.

This process, outlined in Figure 1, gave us a set of 2,418 drugs and drug labels for UniTox. Then, we applied our LLM framework to the UniTox drugs for eight types of toxicity: cardiotoxicity, liver toxicity, renal toxicity, pulmonary toxicity, hematological toxicity, dermatological toxicity, ototoxicity, and infertility. These toxicities were chosen in consultation with our clinician co-authors. The criteria was to include a broad range of organ systems where clinicians would most want standardized toxicity information.

## 2.2 Generating toxicity ratings with LLMs

To generate toxicity ratings from a drug label, we utilized an LLM and chain-of-thought [25] reasoning with a two-tiered prompt system. The first prompt—the "summary prompt"—asks the LLM to read the drug label and summarize the drug's toxicity for a given type of toxicity (e.g., cardiotoxicity). The second prompt—the "rating prompt"—asks the LLM to use only this toxicity summary to produce a toxicity rating, which is either a ternary rating (No, Less, or Most toxicity) or a binary rating (No or Yes toxicity). This ternary prompt allows the model to separate potential "borderline" cases of mild or very rare adverse reactions from more "clear cut" cases of either significant toxicity or no risk of toxicity. If performing well, the model will classify drugs as "Less" toxic if reasonable readers may disagree about whether a rare or mild drug reaction rises to the level of "Toxicity." As a result, we expected the model to have its worst accuracy (compared to clinician validations) on this predicted "Less" category, and better accuracy on the "No" and "Most" categories. The specific prompts provided to the model are below, where `<toxicity type>` is replaced with the toxicity.

---

**Summary Prompt**

Provide a summary of all the parts of the drug label that discuss `<toxicity type>` risks and `<toxicity type>` reactions for this drug. In your summary of each sentence, clearly state whether the drug itself was associated with or caused the `<toxicity type>` risk.

---

**Rating Prompt – Ternary**

Given the above information about a drug, answer "was this drug associated with No `<toxicity type>`, Less `<toxicity type>`, or Most `<toxicity type>`?" Now, answer with just one word: No, Less or Most.

---

**Rating Prompt – Binary**

Given the above information about a drug, answer "was this drug associated with `<toxicity type>`?" Now, answer with just one word: Yes or No.

---

Table 1: Example UniTox Entries

| Generic Name | ABALOPARATIDE | ABEMACICLIB |
|---|---|---|
| Toxicity | Pulmonary | Pulmonary |
| Ternary Rating | No | Most |
| Binary Rating | No | Yes |
| Summary (Trimmed) | ...The sections of the label that detail adverse reactions, warnings, and precautions do not mention any pulmonary-related issues directly associated... | ... associated with significant pulmonary toxicity risks, including severe, life-threatening, or fatal interstitial lung disease (ILD) or pneumonitis... observed in clinical trials, postmarketing settings... |

## 2.3 Validation on DICTrank, DILIrank, and DIRIL

We first validated the toxicity ratings in UniTox by measuring the concordance of these ratings with human-annotated toxicity ratings from three FDA datasets: DICTrank, DILIrank, and DIRIL. This required matching the drugs in UniTox to those in the FDA datasets using the drug data available in these datasets. For DICTrank, we matched by generic drug name. For DILIrank, we used the RxNorm [26] database to pull Structured Product Labeling (SPL) Set IDs for each drug, then matched to the SPL IDs we used. For DIRIL, we matched to our toxicity ratings using moiety UNII codes. Then, for each of these three datasets, we evaluated UniTox and human toxicity rating concordance among the matched drugs. Furthermore, to better understand what drives the LLM's performance, we performed ablations on DICTrank in Section 3.2.1, including a longer prompt with the specific cardiotoxic keywords from DICTrank, using GPT-3.5 instead of GPT-4o, and removing the chain-of-thought step.

## 2.4 Clinician validation on other toxicities

For the five remaining toxicity types without pre-existing validation data, we worked with clinicians to manually validate a subset of the UniTox toxicity ratings. Specifically, we asked clinicians to read the toxicity summary and use both the summary and their knowledge of the drug to validate the toxicity ratings for 200 randomly sampled drugs for each of the five toxicity types (two clinicians, 100 drugs per clinician per toxicity type). For each drug and toxicity type, the clinicians separately evaluated both the ternary and binary toxicity ratings on a scale of 1 to 3, where 1 means "The model's score is factually correct and I agree with it", 2 means "The model's score is reasonable but I don't necessarily agree with it", and 3 means "The model's score is factually incorrect and I disagree." We also asked clinicians to flag if the LLM-generated toxicity summary did not concord with their understanding of a drug and its use.

## 3 Results

Here, we present details about the UniTox dataset (Section 3.1). Then, we discuss our validations on external datasets and the effect of ablations on performance (Section 3.2). Next, we show results of our clinician review of the five toxicities without pre-existing FDA validation data (Section 3.3). Finally, we illustrate the benefit of a unified toxicity dataset by benchmarking several molecular property prediction models on UniTox (Section 3.4).

## 3.1 UniTox

UniTox contains 2,418 drugs with eight types of toxicities. For each drug and toxicity type, UniTox includes (1) a GPT-4o generated summary of the drug label's discussion of that toxicity, (2) a ternary classification into No Toxicity, Less Toxicity, or Most Toxicity, (3) a binary classification into No Toxicity and Yes Toxicity, and (4) the Stuctured Product Labeling (SPL) ID for the document used to generate all data. Properties 1-3 are listed for two examples in Table 1. A key contribution of UniTox is its summaries, which capture the nuance of each drug's toxicity in a fraction of the length of the full text drug labels (297 words on average in the summary compared to 8,254 words on

average in the full drug label). The value also lies in the toxicity ratings, which can be used as "ground-truth" for supervised training of downstream toxicity predictors. Where users wish to modify our LLM-generated ratings, they can utilize the short summaries and avoid reading full-text drug labels.

UniTox is, to the best of our knowledge, the largest human *in vivo* drug-induced toxicity database by number of drugs and number of toxicities. It covers a diverse range of drugs and clinical toxicities that can often be difficult to identify in pre-clinical studies. Figure 2 shows the number of drugs in UniTox with each ternary toxicity rating for each toxicity type. While most toxicity types have a balance of toxic and non-toxic drugs (20–46% classified as Most Toxic), it is worth noting that dermatological toxicity and ototoxicity are outliers with 62% and 4% of the drugs predicted as Most Toxic, respectively.

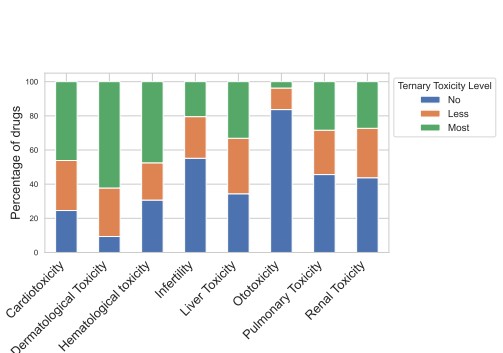

Figure 2: Distribution of ternary toxicity ratings in UniTox across 2,418 drugs.

Figure 3: Heatmap of correlations between different toxicity types in UniTox.

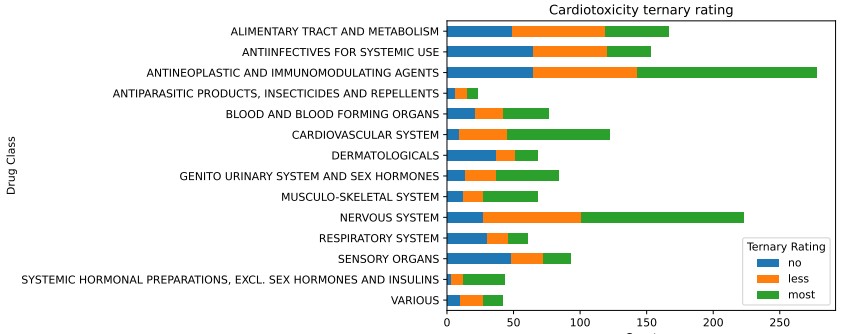

Figure 4: Predicted cardiotoxicity by top-level drug class from WHO-ATC classifications

### 3.1.1 Cross-toxicity analysis and drug class analysis

One of the advantages of a unified toxicity dataset is the ability to determine whether drugs exhibit multiple toxicities. This is, to the best of our knowledge, the largest systematic analysis of how drug toxicities are related. Interestingly, we find the number of binary toxicity ratings per drug approximates a normal distribution, centered at four of the eight toxicities (Figure 7 in Appendix).

We then calculated pairwise correlations across the toxicities, using our binary ratings (Figure 3). We find that liver toxicity and hematological toxicity are the most highly correlated, at 0.45, with pulmonary and cardiotoxicity the second most correlated at 0.30, and liver toxicity and renal toxicity third most correlated at 0.29. We did not find any negative correlations. We believe these results can help future researchers better understand drug toxicity by examining potential causes of these correlations and specific drugs that exhibit unusual patterns of toxicity across systems.

A second advantage of a unified toxicity dataset is the ability to understand toxicity within and across drug classes. To do so, we matched our toxicity predictions with the WHO Anatomical Therapeutic Chemical (ATC) classifications (Figure 4) [27] [28]. We matched on generic name, finding a mapping for 1,501 drugs in UniTox. It is important to note that all drugs in UniTox are FDA-approved. As a result, the difference across classes likely reflects a difference in FDA risk tolerance for different diseases. For example, as expected, immunomodulators and oncology drugs are more likely to be toxic than say, dermatological (e.g., anti-fungal) drugs. This is likely because the FDA is willing to tolerate more toxicity in drugs for potentially fatal diseases than in drugs for low-risk diseases. Still, the analysis allows UniTox users to filter by class or target and understand differences within and between classes.

## 3.2 Validation on external datasets

### 3.2.1 DICTrank

UniTox has 1,181 drug label matches with the DICTrank dataset of 1,318 drugs. Usually, a lack of a match indicates the drug has been withdrawn or discontinued and so a drug label is no longer available. To binarize our results, we consider "Ambiguous-DICT-Concern", "Less-DICT-Concern", and "Most-DICT-Concern" to be toxic, and "No-DICT-Concern" to be non-toxic. Binarizing "Less-DICT-Concern" to toxic is similar to other papers [29], and "Ambiguous-DICT-Concern" makes up only a small (8%) share of matches. We similarly binarized our ternary ratings by combining "Less" and "Most" into a single toxic category. This is the **Ternary** column in Table 2. We also show results from dropping drugs the model predicts are "Less" toxic to focus on the model's clear-cut predictions, in the **Ternary on Predicted No/Most** column. Finally, to compare directly with askFDALabel, we show the results of our binary ratings on only the ground truth "No-DICT-Concern" and "Most-DICT-Concern" subset (**Binary on Ground Truth No/Most** column). Here, we obtain a significantly improved 93.0% accuracy compared to askFDALabel's 77.7% accuracy with a fine-tuned LLM and 71.5% with GPT-3.5.

### 3.2.2 DICTrank ablations and sensitivity analysis

To better understand our performance on the DICTrank dataset, we consider a series of ablations (Table 2). First, we used a keyword summary prompt that contained the full list of DICTrank keywords (e.g., myocardial infarction and Torsade de Pointes). We did not alter the ratings prompts. Performance increases or decreases slightly on our different cuts of the data, likely demonstrating that the GPT-4o model has a strong and accurate internal definition of cardiotoxicity.

Second, we ablated the chain-of-thought step (i.e., the summary prompt), instead providing the full text of the drug label to the model and using just the ternary and binary ratings prompts. We note a consistent decrease in performance. Considering that our full pipeline's rating step considered only the GPT-4o-generated toxicity summaries, this shows the benefit of providing focused and thoughtful information about toxicity.

Third, we switched to GPT-3.5, which required truncating a small number of drug labels to fit into context. askFDALabel achieved a DICTrank accuracy of 71.5% using GPT-3.5, while our prompting strategy with GPT-3.5 achieved 88.4% accuracy. In particular, our prompt specifically asked about "cardiotoxicity" while askFDALabel asked about "cardio-related adverse events or risks", which likely boosted our performance. GPT-3.5 consistently performs worse than GPT-4o.

Finally, to better understand the role of using the full drug label, we applied our ratings prompts on the RAG-retrieved label fragments from askFDALabel. We had access to only the drug label fragments retrieved by the askFDALabel model, rather than the model itself. As a result, we can only compare to the No/Most subset. Both approaches performed similarly. However, generating GPT-4o summaries only involved designing a short prompt, compared to building a custom RAG system, and should return all relevant information in the drug label. The RAG system returns only the top-k fragments, so it may miss vital details. For example, our summary of the full drug label of voclosporin (below) discussed the risk of QT prolongation in sections of the drug label that were not returned by the RAG system. As a result, only the prediction based on the full label was correct.

| DICTrank accuracy (%) | Ternary (n=1181) | Ternary on Predicted No/Most (n) | Binary on Ground Truth No/Most (n) |
|---|---|---|---|
| **Full pipeline** | 84.6 | 92.5 (761) | **93.0** (603) |
| Keyword summary prompt | 88.1 | 90.4 (924) | 93.7 (603) |
| No CoT | 77.8 | 67.4 (629) | 92.5 (603) |
| GPT-3.5 | 77.6 | 77.8 (855) | 88.4 (603) |
| RAG fragment context | | | 94.2 (584) |
| **askFDALabel (previous SOTA)** | | | **77.7** (584) |
| DILIrank accuracy (%) | Ternary (n=819) | Ternary on Predicted No/Most (n=525) | Binary on Ground Truth No/Most (n=363) |
| Full pipeline | 81.1 | 85.0 | 86.2 |
| DIRIL accuracy (%) | Ternary (n=269) | Ternary on Predicted No/Most (n=177) | Binary on Ground Truth No/Most (n=269) |
| Full pipeline | 71.3 | 76.8 | 72.9 |

Table 2: **Validation results across cardiotoxicity (DICTrank), liver toxicity (DILIrank), and renal toxicity (DIRIL) comparing our predictions to expert ratings from the FDA.** We show accuracy using our ternary prompt on the full dataset. We also show accuracy after filtering to only include predicted "No" or "Most" drugs to allow the model to set aside borderline cases. Finally, we show accuracy using our binary prompt after filtering to only include ground-truth "No" or "Most" drugs, allowing for apples-to-apples comparisons with askFDALabel. Our full pipeline on DICTrank significantly outperforms the previous state-of-the-art (askFDALabel). In our ablations, adding additional cardiotoxicity keywords into our summary prompt had an uneven effect on accuracy. Removing the Chain-of-Thought step and moving to GPT-3.5 consistently hurt accuracy on DICTrank. Running our binary prompt on just the fragments of the drug label returned by the FDA's RAG system slightly outperforms using the full drug label, perhaps by limiting extraneous information. Because we only have the FDA's RAG fragments for the ground-truth No/Most subset of DICTrank, we cannot compare to results on the full dataset. Finally, we achieve similarly high accuracy on DILIrank, but our predictions perform worse on DIRIL perhaps due to a differing methodology.

> **Voclosporin:** GPT-4o cardiotoxicity summary | DICTrank: Toxic | GPT-4o: Toxic
>
> ...
> **4. Medication Guide, What are the possible side effects of LUPKYNIS? (Section 17)**:
> **Summary**: LUPKYNIS may cause serious side effects, including a serious heart rhythm problem (QT prolongation).
> **Association**: The drug itself is associated with the risk of a serious heart rhythm problem (QT prolongation).
> In summary, LUPKYNIS is associated with cardiotoxicity risks, specifically QT prolongation, which is dose-dependent and can be clinically significant when combined with other QT-prolonging drugs. This risk is highlighted in multiple sections of the drug label, indicating that the drug itself can cause these cardiotoxic effects.

Most interestingly, our GPT-4o-generated summaries can reason through difficult edge cases. For example, linagliptin is rated as Most-DICT-Concern for cardiotoxicity in DICTrank as the drug label discusses heart failure. However, the drug label notes that "Heart failure has been observed with two other members of the DPP-4 inhibitor class." GPT-4o, when looking only at the three results returned by the RAG system, predicts Yes for cardiotoxicity. However, GPT-4o provided with the full drug label notes linagliptin itself has not been associated with heart failure and discusses two large-scale trials showing linagliptin did not increase major adverse cardiovascular events (see box below). As a result, it predicts linagliptin has No Toxicity in the binary case. While this differs from the ground truth in DICTrank, it is a reasonable response that comprehensively reasons across the drug label.

> **Linagliptin:** GPT-4o cardiotoxicity summary | DICTrank: Toxic | GPT-4o: Non-Toxic
>
> ...
> **Observation**: Heart failure has been observed with two other members of the DPP-4 inhibitor class.
> **Association** It is not explicitly stated that TRADJENTA itself causes heart failure, but the risk is considered due to the class effect.
> ...
> **CARMELINA Trial**: **Primary Endpoint**: The primary endpoint was the time to first occurrence of major adverse cardiovascular events (MACE), including cardiovascular death, non-fatal myocardial infarction, or non-fatal stroke. **Results**: The estimated hazard ratio for MACE associated with TRADJENTA relative to placebo was 1.02, indicating no significant increase in risk. **Conclusion**: TRADJENTA did not show an increased risk of major adverse cardiovascular events compared to placebo.
> ...
> **Conclusion:** **Heart Failure**: While heart failure has been observed with other DPP-4 inhibitors, TRADJENTA itself is not explicitly stated to cause heart failure but should be used with caution in patients with risk factors. **Cardiovascular Events**: Clinical trials (CARMELINA and CAROLINA) indicate that TRADJENTA does not increase the risk of major adverse cardiovascular events compared to placebo or glimepiride.
> ...

### 3.2.3 DILIrank and DIRIL

We performed similar validations of DILIrank (liver toxicity) and DIRIL (renal toxicity), as seen in Table 2. For DILIrank, we achieve similar performance as DICTrank on the 819 drugs where we had a match. Our DIRIL results are less impressive, which may be due to the fact that DIRIL was constructed using a different methodology than DICTrank and DILIrank, in which they primarily took ratings from two previous papers instead of analyzing FDA drug labels for every drug. We note that for 9 of our 25 false positives and for 8 of our 48 false negatives, at least one of the previous papers agreed with GPT-4o's rating rather than the FDA paper's determination.

However, it is also likely that GPT-4o's internal definition of renal toxicity is less calibrated to the FDA's definition than for other toxicities. Given that false negatives were more frequent than false positives, we analyzed the false negatives and found that the GPT-4o-generated summary often noted that the use of the drug is cautioned in renally impaired patients. When binarizing this summary, GPT-4o predicted No Toxicity in these cases. However, the FDA reviewers likely viewed this as a sign of toxicity. This shows that, even when the binary rating of the model may differ from human ratings, GPT-4o condenses valuable information for human reviewers in its summaries.

### 3.3 Clinician evaluation of toxicity ratings

Figure 5 shows the distribution of clinician-derived evaluations of the LLM-generated UniTox ratings (ternary rating). Depending on the toxicity, 85-96% of the drugs were considered accurately rated, 3-12% were ambiguous, and 1-7% were rated incorrectly.

While at least 85% of the clinician scores agreed with the UniTox ratings, the disagreements reveal some edge cases. For many of the drugs where the clinicians gave a rating of 2, the explanation was a lack of direct data or evidence in humans for the specific toxicity. For example, trilaciclib received a UniTox rating of Most Toxicity based on evidence that it may impair fertility in animals; however, the clinician scored this rating with a 2 due to the lack of *human* evidence. On the other hand, trientine hydrochloride capsules received a UniTox rating of No Toxicity as the drug label provided zero evidence that this drug is associated with fertility risks; this also received a clinician score of 2 as the drug label simply did not provide any data about fertility risks and the model was conflating a lack of evidence about toxicity with evidence for a lack of toxicity. Indeed, based on this feedback, we added a new prompt for infertility to clarify the level of available evidence (added as an additional column in UniTox).

Sometimes, clinicians' disagreements with the model highlight genuine errors in the model's assessment. For example, ganciclovir injection received a UniTox rating of Less Toxicity for ototoxicity.

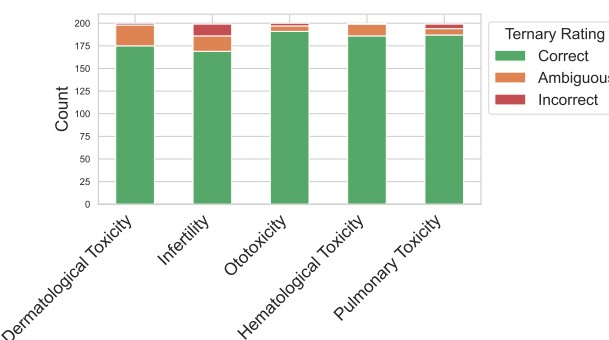

Figure 5: Distribution of clinician validation scores on GPT-4o-generated ternary toxicity ratings.

However, the drug label lists "tinnitus, ear pain, deafness" as observed adverse effects, which should clearly be considered Most Toxicity.

### 3.4 Benchmarking toxicity prediction models

Next, we demonstrate the utility of UniTox by using it to train machine learning models to predict toxicity from molecular structure, which is an important aspect of drug discovery. We benchmarked four machine learning model architectures: (1) Chemprop [19], a widely-used graph neural network (GNN) for molecular property prediction, (2) Multilayer Perceptron (MLP), (3) Random Forest (RF), and (4) Support Vector Machine (SVM). For each machine learning model architecture, we experimented with two methods for featurizing the input molecules: (1) RDKit fingerprints, which consist of 200 molecular features (e.g., formal charge of the molecule) computed by RDKit [30], and (2) Morgan fingerprints, which are binary vectors indicating the presence or absence of small neighborhoods of atoms and bonds in the molecule. The MLP, RF, and SVM models require one of these input featurizations, while the Chemprop model can either be run as a pure GNN operating on the atoms and bonds of the molecule or as a GNN augmented with one of the featurizations as additional input.

For each model, we performed ten-fold cross-validation using a challenging scaffold split, which means that molecules were clustered by their core molecular scaffold and clusters were placed either entirely in the train set or entirely in the test set. This ensures that similar drugs do not leak between train and test. The Chemprop and MLP models were trained in a multi-task setting with one model predicting all eight toxicities, while separate RF and SVM models were trained for each toxicity type due to architectural limitations.

Since these models are only designed to work with small molecules, we restricted UniTox to the set of small molecule drugs (e.g., excluding biologics). We then used the PubChem [31] API to match generic drug names to SMILES. We deduplicated drugs by SMILES and removed any SMILES where at least one of the toxicity ratings across the eight toxicities differ between different drugs with the same SMILES (e.g., different formulations of the same drug). This resulted in a deduplicated set of 1,349 drugs with unique SMILES and concordant toxicity ratings, which we refer to as the UniTox Small Molecule Benchmark subset. We trained our models on the binary task of predicting No Toxicity or Most Toxicity (ignoring Less Toxicity) from the ternary ratings. We note that parameters such as dosage would not affect performance as there is only a single FDA drug label across all approved clinical dosages.

As shown in Figure 6, all models perform reasonably well given the dataset size and the inherent biological complexity of human *in vivo* toxicity. Their performance is generally within the range of other molecular property prediction models in the literature (e.g., ADMET-AI [20]). Some models, such as Chemprop and Chemprop RDKit, perform poorly on dermatological toxicity and ototoxicity, perhaps in part due to the extreme class imbalance present in both datasets (62% Most Toxicity with dermatological toxicity and 4% Most Toxicity with ototoxicity in UniTox Small Molecule Benchmark). Overall, these results illustrate the benefit of building comprehensive toxicity datasets as it enables training molecular property prediction models that can generalize to new molecules and could potentially be used as *in silico* toxicity screening tools prior to clinical validation.

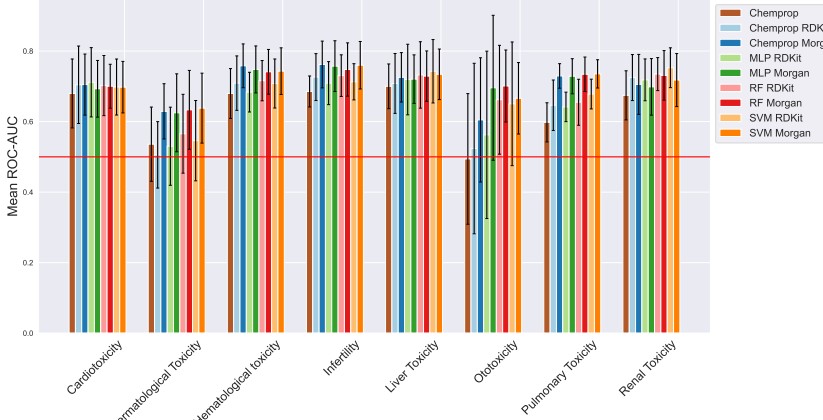

Figure 6: Performance of several machine learning models trained on the UniTox Small Molecule Benchmark subset to predict the No/Most ternary ratings (mean ± standard deviation ROC-AUC across ten-fold cross-validation).

## 4 Discussion

In this work, we demonstrated the ability of GPT-4o to rapidly generate useful and accurate summaries of complex drug labels. When binarized, these summaries had high concordance with the external DICTrank (cardiotoxicity) and DILIrank (liver toxicity) datasets, and to a lesser extent, to the DIRIL (renal toxicity) dataset. UniTox also had a high concordance with clinical reviewers for toxicities without pre-existing comparable quantitative validation data. We demonstrate the value of these summaries, and their binarized values, by training molecular classifiers with predictive value. These ratings, even where occasionally noisy, can serve as a benchmark for future classifiers that seek to demonstrate consistent performance across toxicities. Such consistent evaluation of downstream classifiers was not previously possible. Finally, we provide insight into the co-occurrence of multiple toxicities from drugs in a unified format not previously available.

The clearest limitation of our work is the challenge of going from a nuanced summary of the drug label to a binary or ternary rating. We note several common challenges for rating toxicity in cases where (1) toxicity occurred only in specific or pre-disposed populations (e.g., children or impaired patients), (2) toxicity occurred only in other drugs of the same class, only in animals, or only at high doses that may exceed clinical relevance, (3) there were common but mild reactions (e.g., rashes for dermatological toxicity), and (4) reactions may occur only when specifically studied (e.g., infertility), so a lack of evidence may not be sufficient to conclude a lack of toxicity. These circumstances were often discussed in detail in GPT-4o's generated summaries but were lost in the binary or ternary ratings. While we preferred simple prompts, perhaps more complicated ratings prompts could better handle these. For example, future work could set a higher bar for dermatological toxicity.

There are several additional limitations. First, we only have these detailed drug labels for FDA-approved drugs, so we cannot apply the same methodology to failed drugs or trials. This could limit the potential of our molecular classifiers by limiting the diversity of molecules they are trained on. Second, because we wanted to train a unified model, we are not able to train a classifier across small molecules and biologics. We focused on models that use SMILES codes and a limited set of additional features to predict small molecule toxicity. Finally, while we have validated our approach by comparing to existing FDA datasets and our own clinician review of 200 drugs, future work could consider additional or larger validation approaches.

We want to note the ethical importance of accuracy in this application area. We have taken steps to validate our predictions, and we provide the nuanced GPT-4o summaries based on drug labels. Still, we note here that these are LLM-generated predictions intended for drug research; they are not medical advice and are not meant to inform healthcare decisions.

As LLMs are used in more information extraction tasks, it is important to understand their strengths and limitations. We demonstrate their value by creating an accurate and useful dataset in a fraction of the time it would take humans to process this amount of text. In particular, we show LLMs' ability to summarize text while maintaining its key information and nuance, and we create useful drug ratings for downstream classifiers. Still, we highlight when and how further condensing that information into a single word remains challenging, for both models and humans.

## 5    Acknowledgments

The authors would like to thank Weida Tong and Leihong Wu of the FDA's Division of Bionformatics and Biostatistics at the National Center for Toxicological Research. JS is supported by the Arc Institute. KS is supported by the Knight-Hennessy Fellowship and the Bio-X Fellowship. ES is supported by NSF GRFP grant DGE-2146755.

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

# A    Appendix

## A.1    Negative and Positive Predictive Values

To demonstrate that our key results were not significantly imbalanced, we include Positive Predictive Value (PPV) and Negative Predictive Value (NPV) across our full pipeline models for Cardiotoxicity, Liver Toxicity, and Renal Toxicity in Table 3.

Table 3: DICTrank, DILIrank, and DIRIL Negative Predictive Value (NPV) and Positive Predictive Value (PPV)

| DICTrank | Ternary (n=1181) | Ternary on Predicted No/Most (n=761) | Binary on Ground Truth No/Most (n=603) |
|---|---|---|---|
| NPV (%) | 79.6 | 79.7 | 96.2 |
| PPV (%) | 85.7 | 97.1 | 90.0 |
| DILIrank | Ternary (n=819) | Ternary on Predicted No/Most (n=525) | Binary on Ground Truth No/Most (n=363) |
| NPV (%) | 72.4 | 72.4 | 97.5 |
| PPV (%) | 83.8 | 92.4 | 71.9 |
| DIRIL | Ternary (n=269) | Ternary on Predicted No/Most (n=177) | Binary on Ground Truth No/Most (n=269) |
| NPV (%) | 75.9 | 75.9 | 66.4 |
| PPV (%) | 69.4 | 77.7 | 80.2 |

## A.2    Histogram of toxicities

As part of the cross-toxicity analysis, we found that the number of toxicities per drug appeared to generally approximate a normal distribution. We present that result here in Figure 7.

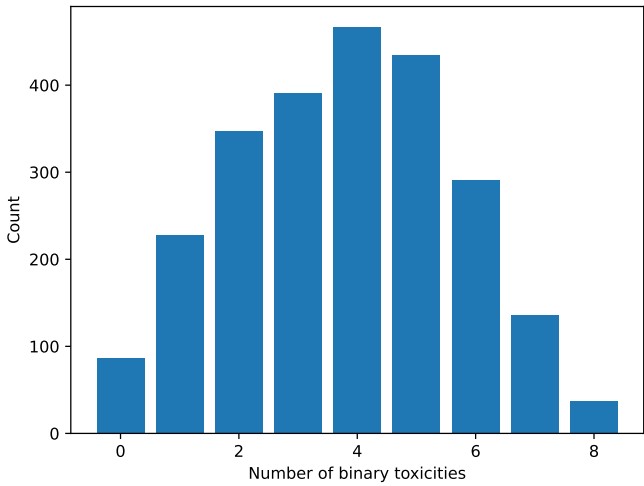

Figure 7: Histogram of toxicities per drug, from the binary toxicity ratings

## A.3    Statistical Significance

To evaluate the potential statistical significance of our clinical validations, we ran a series of permutation tests. To do so, we first convert our clinician ratings into classifications for each drug. If the clinician agreed with the model, we treat it as the same binary rating. If the clinician slightly or

significantly disagreed, we assume that the clinician gave the opposite rating. We then shuffle the LLM-generated classifications. This represents the null hypothesis that these classifications have zero predictive value for the true rating. We calculate the chance of getting our observed level of agreement between clinician and LLM classifications. Based on 1,000 permutations, we find a p-value <0.001 for each toxicity, and reject this null hypothesis, as shown in Figure 8.

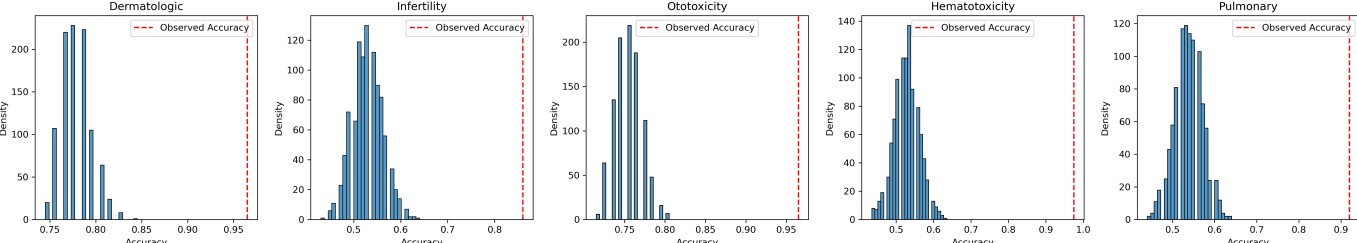

Figure 8: Permutation tests evaluating agreement between clinician and LLM ratings of toxicity across 1,000 simulations.

## A.4 Evaluator Instructions

Finally, we provide the instructions given to our clinical validators in the box below. The individual evaluations per drug in UniTox (200 randomly selected samples) is available on our GitHub.

> ### Clinician evaluator instructions
>
> Each of these spreadsheets has 5 tabs corresponding to the toxicities that do not have external validation sets (Dermatological toxicity, Hematological, Infertility, Ototoxicity, Pulmonary toxicity). Here is a breakdown of the columns in each sheet:
>
> - "generic_name": Drug name
> - "reasoning": Model's reasoning for why it gave the determinations it did
> - "less_determination": Model's determination of risk on a scale of (No risk, Less risk, Most risk)
> - "binary_determination": Model's determination of risk on a binary scale (Yes / No)
> - "less_determination_score": Empty column for you to fill out scoring whether you agree with the "No/Less/Most" determination or not
> - "binary_determination_score": Empty column for you to fill out scoring whether you agree with the "Binary" determination or not
> - "issues_with_summary" column to briefly jot down if something seems truly hallucinated or very different from your prior knowledge about one of these drugs.
>
> When specifying your scores, please score them 1-3 where:
>
> - 1 = The model's score is factually correct and I agree with it
> - 2 = The model's score is reasonable but I don't necessarily agree with it (this is the gray-zone)
> - 3 = The model's score is factually incorrect and I disagree

