# UniTox: Supplementary Materials

## Motivation

**For what purpose was the dataset created?**
UniTox was created as a unified toxicity dataset across eight types of drug toxicities (cardiotoxicity, liver toxicity, renal toxicity, pulmonary toxicity, hematological toxicity, dermatological toxicity, ototoxicity, and infertility). We generated information across all toxicities for the same set of 2,418 drugs with the same methodology of applying LLMs. For each drug, for each toxicity, we provide an LLM-generated summary of the relevant portions of the drug label, as well as ternary (No/Less/Most) predictions and binary (No/Yes) predictions for that toxicity.

**Who created the dataset (e.g., which team, research group) and on behalf of which entity (e.g., company, institution, organization)?**
The dataset was created by Jake Silberg, Kyle Swanson, Elana Simon, Angela Zhang, Zaniar Ghazizadeh, and James Zou at Stanford University, as well as Scott Ogden and Hisham Hamadeh at GenMab.

**Who funded the creation of the dataset?**
The Chan-Zuckerberg Biohub

## Composition

**What do the instances that comprise the dataset represent?**
Each dataset instance is a single drug. For each drug, we provide information across eight toxicities, as well as a unique identifier in Structured Drug Labeling (SPL) format for the drug label used to create the toxicity information.

**How many instances are there in total?**
There are 2,418 drugs in the dataset and each drug has information on eight toxicities.

**Does the dataset contain all possible instances or is it a sample (not necessarily random) of instances from a larger set?**

The dataset is a subset of all possible NDA, ANDA, and BLA drug labels for FDA approved drugs (50,617 labels in total). We de-duplicated these drugs by unique generic names. Drugs that do not have a current FDA-approved label (e.g., withdrawn or discontinued drugs) are not included.

**What data does each instance consist of?**
Each instance is a single drug. For each instance, there are eight toxicities, and for each toxicity, there is an LLM-generated summary of the relevant sections of the drug label, a ternary prediction (No/Less/Most), and a binary prediction (No/Yes). Each instance also provides the unique SPL ID, allowing users to find the exact text used to generate the instance data.

**Is there a label or target associated with each instance?**
Each instance has LLM-generated toxicity labels, both in ternary (No/Less/Most) and binary (No/Yes) form, for eight toxicity types.

**Is any information missing from individual instances?**
All instances have a generic drug name, SPL ID, and LLM-generated toxicity summaries and labels. However, not all instances have SMILES. Only drugs that are small molecules whose generic name matches an entry in PubChem have a SMILES.

**Are relationships between individual instances made explicit (e.g., users' movie ratings, social network links)?**
No, there are no relationships between instances (e.g., drug classes or disease treatment classes) made explicit.

**Are there recommended data splits (e.g., training, development/validation, testing)?**
There are not suggested data splits.

**Are there any errors, sources of noise, or redundancies in the dataset?**
There are potential redundancies of the following forms:

1) Because we de-duplicated drugs based on generic name, drugs with the same moiety may appear with different names (e.g., Abacavir and Abacavir Sulfate)
2) Any typos or inconsistencies in a drug name could cause it to appear multiple times in the dataset

**Is the dataset self-contained, or does it link to or otherwise rely on external resources (e.g., websites, tweets, other datasets)?**
The dataset is self-contained except for the FDA labels, which were used by the LLM to generate the toxicity summaries and labels but are not included directly in the dataset. The FDA labels can be found on the FDALabel website (https://www.fda.gov/science-research/bioinformatics-tools/fdalabel-full-text-search-drug-product-labeling) based on the SPL IDs included in the dataset.

**Does the dataset contain data that might be considered confidential (e.g., data that is protected by legal privilege or by doctor–patient confidentiality, data that includes the content of individuals' nonpublic communications)?**
No, all data in the dataset, and all data used to generate the dataset are publicly available data published in several forms on FDA websites (https://www.fda.gov/about-fda/about-website/website-policies).

**Who was involved in the data collection process and how were they compensated?**
These labels are created by the FDA in discussion and consultation with the drugmaker. Our LLM-generated summaries involved no data collection by anyone other than the authors.

**Over what timeframe was the data collected?**
We have not identified the oldest label still in the dataset. Labels are updated regularly as new information about a drug becomes available.

**Were any ethical review processes conducted?**

Ethical reviews of the LLM-processing steps were not conducted as the risk was considered minimal

## Preprocessing/cleaning/labeling

**Was any preprocessing/cleaning/labeling of the data done?**
The deduplication process has been described above. The HTML drug label was stripped of tags using the Beautiful Soup python package. Figures in the drug label were not processed or considered.

**Was the "raw" data saved in addition to the preprocessed/cleaned/labeled data?**
The exact drug label text used to generate our summaries and predictions can be identified from the SPL ID. Additionally, the raw query results from FDALabel, prior to deduplication are available in our github (https://github.com/jsilbergDS/UniTox).

**Is the software that was used to preprocess/clean/label the data available?**
All the code used for deduplication is available from our github (https://github.com/jsilbergDS/UniTox)

## Uses

**Has the dataset been used for any tasks already?**
Yes, in the paper that published this dataset, a subset of the data was used to train several machine learning classifiers to predict small molecule drug toxicities.

**Is there a repository that links to any or all papers or systems that use the dataset?**
No, there is no central repository for all papers using this dataset.

**What (other) tasks could the dataset be used for?**
This data could be used for other tasks related to predicting drug toxicity or understanding the relations between different types of toxicity.

**Is there anything about the composition of the dataset or the way it was collected and preprocessed/cleaned/labeled that might impact future uses?**
It is worth noting that the toxicity summaries and labels are LLM-generated and therefore are not always accurate, though we have taken some measures to validate the summaries.

**Are there tasks for which the dataset should not be used?**
The dataset should not be used for patients to decide whether a drug is safe to take. Patients should always consult medical experts about these drugs.

## Distribution

**Will the dataset be distributed to third parties outside of the entity (e.g., company, institution, organization) on behalf of which the dataset was created?**
Yes, the dataset is publicly available on the internet.

**How will the dataset be distributed?**
The dataset is on Zenodo with a DOI.

**When will the dataset be distributed?**
The dataset was first distributed in June 2024.

**Will the dataset be distributed under a copyright or other intellectual property (IP) license, and/or under applicable terms of use (ToU)?**
The dataset is distributed under a CC BY 4.0 license.

**Have any third parties imposed IP-based or other restrictions on the data associated with the instances?**
No, since the FDA drug labels from which the dataset was generated are in the public domain.

**Do any export controls or other regulatory restrictions apply to the dataset or to individual instances?**
No.

## Maintenance

**Who will be supporting/hosting/maintaining the dataset?**
The Zou lab at Stanford University will be supporting and hosting UniTox.

**How can the owner/curator/manager of the dataset be contacted (e.g., email address)?**
Jake Silberg can be contacted at jsilberg at stanford.edu

**Is there an erratum?**
This will be posted on the dataset webpage.

**If the dataset relates to people, are there applicable limits on the retention of the data associated with the instances (e.g., were the individuals in question told that their data would be retained for a fixed period of time and then deleted)?**
N/A

**Will older versions of the dataset continue to be supported/hosted/maintained?**
To avoid providing inconsistent drug information, we do not anticipate hosting previous versions, though any errata will be made available

**If others want to extend/augment/build on/contribute to the dataset, is there a mechanism for them to do so?**
We suggest contacting the authors.

**URL to website/hosting**

Zenodo:  https://zenodo.org/records/11627822 (10.5281/zenodo.11627822)

Website: https://zou-group.github.io/UniTox-website/

**Responsibility and licenses**

We, the authors of UniTox, accept responsibility for any violation of rights. Because all underlying data in UniTox is generated from FDA labels, we do not anticipate any such violations, but nonetheless accept responsibility for unanticipated challenges. The UniTox data is licensed with a CC BY 4.0 license and code used to produce the data is licensed with an MIT license.

**Hosting/maintenance plan**
The Zou lab at Stanford University will be supporting and hosting UniTox. In consultation with clinicians, we anticipate regular updates as we continue to validate and improve the LLM-generated summaries and predictions. Updates to the summaries and predictions for existing drugs will result in the release of a new minor version (e.g., 1.1). Updates that result in drugs being added or removed to the dataset will result in the release of a new major version (e.g., 2.0).

**Data format**
The UniTox dataset is a standard CSV file and can be read using programs such as Excel or using coding packages such as Pandas in Python.

**Structured metadata about the dataset**

Structured metadata is available in the Zenodo record for UniTox.