# OpenReview forum: "UniTox: Leveraging LLMs to Curate a Unified Dataset of Drug-Induced Toxicity from FDA Labels"
_NeurIPS.cc/2024/Datasets_and_Benchmarks_Track — NeurIPS 2024 Track Datasets and Benchmarks Spotlight_

### Official Review · Reviewer_LMvU · 2024-07-20
**Valuable LLM based dataset curation, benchmarking, and GNN model development for drug-induced toxicity application**

**Rating:** 9
**Confidence:** 4
**Correctness:** Yes
**Clarity:** Yes

**Review:**

The manuscript is well-written and easy to read, the results are clearly presented, and the work is original. The authors have done a good job in justifying the motivation behind the work and the overall quality of the work is highly innovative. The results are promising and showcase the potential of developing frameworks such as the UniTox for impacting biomedical research. The significance of the research is also evident by the fact that the UniTox framework demonstrates the potential for saving precious time spent in human expert reviews or at least assisting such reviewers to make more informed decisions efficiently. The generated dataset also provides an opportunity to develop new machine learning models to predict toxicity across all types.

Pros: (1) Diverse dataset in terms of number of drugs (2418) and number of toxicities (8) considered, (2) Dataset is uniform in terms of both positive and negative examples for toxicities, (3) Largest human in vivo dataset that accurately reflects drug effects and biological cross-toxicity information, (4) Benchmarking was validated by involving a human in the loop in the form of recruited clinician reviews, (5) Accuracy obtained in studies involving external validation sets and comparison to other models is high, (6) Cross-toxicity analysis, ablation studies, and sensitivity analysis was carried out, (6) Using the dataset, a proof-of-concept application is shown in the form of graph neural network development.

My only con is that only one type of LLM was utilized in the whole study. The dataset is highly LLM dependent so there might be value in finding a way to generalize the label summaries generated by various LLMs to make an even more uniform dataset.

**Strengths:**

Please see above for the Pros.

**Additional Feedback:**

None

**Documentation:**

The GitHub code is openly available.

**Limitations:**

The authors do a really good job of discussing the limitations associated with their work in terms of the complexity associated with converting the text summaries to classification bins. They also provide a disclaimer around the fact that the work is not meant to inform medical advice or make healthcare decisions.

**Opportunities For Improvement:**

Maybe benchmarking the performance of other LLMs than just GPT to check if the results hold and to provide an even diverse set of drug label summaries depending on the model used.

**Relation To Prior Work:**

Yes

**Summary And Contributions:**

The authors present a framework for drug-induced toxicity application where a dataset is created using a large language model by processing 2,418 FDA-approved drug label documents. The dataset is generated to contain short summaries (298 words on average) compared to documents that exceed 100 pages per drug, potentially overcoming time-intensive expert reviewer curation. The provided dataset additionally contains the ternary and binary toxicity ratings for each drug converted from analyzing these short summaries using user-defined prompts. The work also shows the creative use of recruited human clinicians for cross-verifying results and proof-of-concept machine learning model development using the curated data.

---

> ### Author Rebuttal · Authors · 2024-08-16
>
> We thank the reviewer for the support and suggestions. We agree with the benefit of testing our pipeline with other LLMs beyond GPT-4o and comparing the results. We will try to perform this analysis for future work.

---

### Official Review · Reviewer_qLGx · 2024-07-22
**Toxicity dataset based on LLM predictions**

**Rating:** 7
**Confidence:** 3
**Correctness:** N/A

**Review:**

Overall it's an original research that contains a cleaver use of LLMs to propose a novel dataset for an important task. Majority of the paper is clearly written, but some sections, in particular ablation study and associated table, are somehow confusing and could benefit from some adjustments. In my opinion this is significant enough issue in terms or presentation of the results that it made my lower my rating quite a bit, but it would be increased if addressed.

**Strengths:**

- Tackles an important problem of drug toxicity, proposes a valuable dataset contribution.
- Useful ablation on using chain-of-thought step and GPT version.
- Illustrative examples of edge cases and failure modes.
- Evaluation of a GNN trained on the extracted labels.

**Additional Feedback:**

N/A

**Clarity:**

As discussed above, in my opinion the paper could benefit from adjusting some parts of the text.

**Documentation:**

N/A

**Limitations:**

One limitation I find it might be beneficial to discuss is the fact that the dataset uses FDA-approved drugs exclusively, and because of that contains fairly "reasonable" molecules. I wonder if this does not introduce some biases in terms of domain of applicability, and how the predictions of a model trained on such dataset would generalize out of this FDA-approved-drug distribution.

**Opportunities For Improvement:**

- Table 2 is very confusing and could benefit from a better caption and/or restructuring the table. NPV/PPV are also never defined, and it’s not clear which metric is even used for the ablation, as for the other row groups it seems to be the case that every row corresponds to a different metric. Because of the above, it’s not even immediately clear if (and to what extent) specific ablation variants outperform.
- Perhaps I am misunderstanding the RAG-retrieved label ablation, but my understanding was that relevant fragments were taken from the askFDALabel’s LLM, but then the final prediction was done using the same LLM and prompt as in this paper. If that’s the case, it’s not clear why no ternary results are included for it. If that’s not the case, some comment on why it wouldn’t be feasible/reasonable would be appreciated. In either case, it should be more clearly described.
- Perhaps this is obvious for anyone familiar with them, but based on the introduction alone it’s not entirely clear what FDA drug labels *are* (what information they contain). Since as far as I understand the term has a different meaning than what labels typically mean in machine learning, a very short sentence on this might be useful.

**Relation To Prior Work:**

N/A

**Summary And Contributions:**

The authors propose a novel toxicity dataset that contains FDA-approved drugs and assigns them toxicity labels based on the LLM predictions on an associated report.

---

> ### Author Rebuttal · Authors · 2024-08-16
>
> We thank the reviewer for the helpful feedback and suggestions on improving the clarity of our work. We respond here:
> * __Improving Table 2:__ We apologize for the confusion on Negative Predictive Value and Positive Predictive Value, which should have been clearly defined. We have moved these metrics and their definition to a new table in the appendix.
> * We should also have clarified that all our ablation comparisons were to the Accuracy numbers of DICTrank, rather than any other metrics. In the attached PDF, we add a simplified Table 2, and emphasize its clarified caption below, discussing our results and ablations:
>   * __Table 2: Validation results across cardiotoxicity (DICTrank), liver toxicity (DILIrank), and renal toxicity (DIRIL) comparing our predictions to expert ratings from the FDA.__ We show accuracy using our ternary prompt on the full dataset. We also show accuracy after filtering to only include predicted “No" or "Most” drugs to allow the model to set aside borderline cases. Finally, we show accuracy using our binary prompt after filtering to only include ground-truth “No” or “Most” drugs, allowing for apples-to-apples comparisons with askFDALabel. Our full pipeline on DICTrank significantly outperforms the previous state-of-the-art (askFDALabel). In our ablations, adding additional cardiotoxicity keywords into our summary prompt had an uneven effect on accuracy. Removing the Chain-of-Thought step and moving to GPT-3.5 consistently hurt accuracy on DICTrank. Running our binary prompt on just the fragments of the drug label returned by the FDA's RAG system slightly outperforms using the full drug label, perhaps by limiting extraneous information. Because we only have the FDA's RAG fragments for the ground-truth No/Most subset of DICTrank, we cannot compare to results on the full dataset. Finally, we achieve similarly high accuracy on DILIrank, but our predictions perform worse on DIRIL perhaps due to a differing methodology.
> * __Ternary comparisons with askFDALabel:__ We thank the reviewer for this opportunity to clarify. The authors of askFDALabel released the text fragments of drug labels returned by their RAG system, however they have not released their code or the RAG module itself. As a result, for the “RAG fragment context” ablation, we can only run our prompting strategy on drugs where they released their RAG-retrieved fragments. They only used the ground-truth No and Most subset of DICTrank (with a few drugs from this subset missing from the release). As a result, running the ternary prompt on this filtered subset is not apples-to-apples comparable to the other results in the Ternary columns. That said, running the Ternary prompt on this filtered subset yields an accuracy of 88.9%, and accuracy of 92.5% if we filter out the drugs where the LLM predicts “Less” (n=560), though we do not include these in Table 2 to avoid unfair comparisons.
> * __FDA drug label:__ We thank the reviewer for the opportunity to clarify here. We have, as suggested, included a brief description given the confusion with the machine learning “label” term: An FDA Drug Label is a comprehensive regulatory document written in collaboration between the FDA and the pharmaceutical company seeking drug approval. It contains all the information that clinical prescribers and/or patients taking this medicine might want to know, such as Indications and Dosages (what the drug should be taken to treat), Warnings and Precautions (any suspected risks of taking the drug), and a summary of the efficacy and safety results from all Clinical Trials reviewed by the FDA as part of their decision to approve the drug. FDA Drug Labels are typically about 10-20 pages long, but some FDA Drug Label documents can be over 100 pages long. The FDA continuously revises these drugs labels as ongoing benefits and safety risk information about a drug becomes available after the initial drug approval.
> * __Using FDA-approved drugs:__ We agree with the reviewer that for our discussion section, we should further note that because the dataset is based on FDA-approved drugs, this is a limitation on dataset size and potentially molecular diversity.

---

> > ### Comment · Reviewer_qLGx · 2024-08-29
> >
> > Thank you for addressing my comments, since the raised concerns were addressed I increased my score.

---

> > > ### Author Response · Authors · 2024-08-30
> > > **Thank you!**
> > >
> > > We would like to thank the reviewer again for the feedback that has improved our work!

---

### Official Review · Reviewer_ibpw · 2024-07-26
**A Novel Dataset for Drug Toxicity: Incompleteness and Reliability Concerns**

**Rating:** 6
**Confidence:** 4
**Clarity:** The paper is well written and easy to…

**Review:**

### Quality:
The design choices made in the paper are justified, and the proposed framework for data generation is reasonable.

### Clarity:
The paper is clearly written and easy to follow.

### Originality:
Although other toxicity datasets exist, this is the first of its kind for toxicity derived from human in-vivo data.

### Significance, Pros, and Cons:
For a detailed discussion, refer to the sections on "Strengths" and "Opportunities for Improvement."


## **Review Summary:**

This paper presents a novel dataset for drug toxicity, derived from FDA labels of approved drugs and summarized using Large Language Models (LLMs). The dataset is a step forward, addressing a gap in human in-vivo drug toxicity data. Despite its potential to aid in constructing in-silico models, the dataset has some limitations. The benchmarking is limited to small molecules without dosing information, and the human evaluation process lacks detailed reliability metrics and instructions. Additionally, the dataset's small size raises questions about its overall utility and representativeness. Further work is needed to include a broader range of drugs and detailed methodological explanations to enhance the dataset's comprehensiveness and reliability.

**Strengths:**

- **Addresses Critical Issue:** The paper tackles the important problem of drug toxicity, which affects all stages of drug discovery.
- **Novel Dataset:** Introduces a human in-vivo dataset for drug toxicity, filling a significant gap in existing datasets. The dataset could significantly aid in constructing in-silico models for drug toxicity prediction
- **Methodology:** Employs LLMs to summarize and categorize extensive FDA label information effectively.
- **Expert Validation:** A subset of the dataset is validated by human experts, ensuring reliability and accuracy.
- **Benchmarking:** Conducts a benchmark on a portion of the dataset to evaluate its performance, providing initial insights into its utility.

**Additional Feedback:**

None

**Correctness:**

The claims made in the paper are adequately supported. However, the reliability measures provided do not sufficiently assure the overall reliability of the dataset.

**Documentation:**

The dataset is sufficiently documented and is easy to access/use. The dataset is a simple CSV file, and maintenance should be straightforward.

**Limitations:**

The authors discuss the limitations of their work, but not sufficiently. While they address the limitations of the dataset development methodology (i.e., using LLMs) and the condensation of toxicity information into ternary labels, additional limitations should be considered:

- **Size and Representativeness of the Molecule Space:** The dataset only covers a small number of small molecule drugs and biologics, which may hinder the learning of robust ML models. The authors should elaborate on how this gap can be bridged, potentially by supplementing with in-vivo datasets.
- **Inadequate Benchmarking:** The benchmark is conducted on a single GNN model and does not include biologics. Additionally, the model cannot utilize differentiating drug information such as dosage. The authors should acknowledge these deficiencies in the current benchmarks within the paper.

There are no significant potential negative societal impacts from this work.

**Opportunities For Improvement:**

**Limitations/Opportunities for Improvement:**

- **Inadequate Benchmarking:** The dataset is benchmarked only on small molecules, with no dosing information passed to the model. Future work should consider including a broader range of models from the drug representation learning literature and incorporate additional parameters.
- **Questionable Human Evaluation Reliability:** The human evaluation process lacks convincing reliability metrics. The paper should provide overall p-values (e.g., under a hypergeometric test) and clarify that only about 100 samples are annotated.
- **Lack of Detailed Human Annotation Instructions:** Specific instructions given to the human annotators should be included. Additionally, the samples and outcomes of the human evaluation should be detailed.
- **Small Dataset size:** The size of the dataset is limited. Future work should consider including drugs from clinical trials, including failed drugs, to create a larger dataset. Integrating other toxicity datasets, such as in-vivo ones, is also recommended.
- **Uncertain Overall Utility:** The overall utility of the dataset is not adequately known due to its small size. It is unclear if the dataset is representative of the large drug molecule space and whether models can be effectively trained. The inclusion of biologics and the potential models that can be learned should be elaborated.
- **Elaboration on Data Point Categories:** The categories of the data points chosen need further elaboration. The paper should explain how these categories were developed.

**Relation To Prior Work:**

The paper discusses previous related works and connections with the current work.

**Summary And Contributions:**

This paper presents a new dataset for drug toxicity. The dataset is derived from FDA labels of approved drugs. LLMs were employed to summarize the FDA label information, which was then categorized into various toxicity categories. A small subset of the dataset was validated by human experts. Additionally, the paper benchmarks a portion of the dataset to evaluate its performance.

---

> ### Author Rebuttal · Authors · 2024-08-16
>
> We thank the reviewer for their critiques and suggestions. We note our responses here, including the improvements we’ve made in response:
> * __Benchmarking with a broader range of models:__ We thank the reviewer for this suggestion. Our benchmarking now includes our GNN, a support vector machine, and a multilayer perceptron. For each model type, we now also tested augmenting with Morgan fingerprints (binary indicators of substructures) or RDKit fingerprints (physicochemical features). We find that no model type dominates across toxicities. Please see an updated Figure 5 attached.
> * __Including parameters such as dosage:__ Please note that adding dosage information to the model would not affect performance. This is because our “ground truth” ratings are based on processing the FDA Drug Label, which is a single document across all approved drug dosages. For example, consider this FDA label for Keytruda: “Keytruda can cause immune-mediated rash or dermatitis. Exfoliative dermatitis, including Stevens Johnson Syndrome, has occurred.” The FDA notes this risk for the drug rather than any specific dose. As a result, we want our classifiers to learn the risk of molecules across typical clinical doses. We are also following other toxicity datasets. For example, LiverTox, by the National Institute of Diabetes and Digestive and Kidney Diseases, has a rating scale per drug, without dosage.
> * __Human Evaluation Reliability:__ We thank the reviewer for this suggestion. In response, we doubled the number of clinician-reviewed validation samples from 100 to 200 drugs and clarified this in the text. We find similar concordance, from 85% to 96%. Please find the updated Figure 4 attached.
> - __P-values for Human Evaluations:__ We thank the reviewer for this suggestion, In response, we conducted the following experiment:
>   - We convert our clinician ratings into classifications for the drug. If the clinician agreed, we treat it as the same binary rating. If the clinician slightly or significantly disagreed, we consider that the clinician gave the opposite rating.
>   - We then run a permutation test, shuffling the LLM-generated classifications. This represents the null hypothesis that these classifications have zero predictive value. We calculate the chance of getting our observed level of agreement between clinician and LLM classifications. Based on 1000 permutations, we find a p-value <.001 for each toxicity, and reject this null hypothesis.
>   - Finally, we should note that our clinician agreement validation is only one part of our human validation. Table 2 demonstrates the concordance of our LLM-generated classifications on three of our eight toxicities by comparing with three validation datasets created by clinical experts at the FDA. For example, DICTrank contains over 1100 expert-created ratings and DILIrank contains over 800.
> * __Human Evaluator Instructions and Results:__ We thank the reviewer for this suggestion. We have added these instructions and the clinician validation results to an appendix.
> * __Dataset Size and integrating in-vivo datasets:__ We note the challenge of integrating additional in-vivo datasets outside of FDA-approved drugs given that our contribution is a unified methodology across toxicities. Other in-vivo datasets are typically limited to a single organ (e.g., LiverTox), so integrating would lead to different drugs per toxicity and differing methodologies. Similarly, drawing on failed clinical trials would only allow us to add the single toxicity that caused the drug to fail the trial rather than all toxicities. Finally, we note that our methodology exactly matches a proven approach taken by FDA experts in DICTrank, whereas combining various databases may not match the methodology that human experts would prefer.
> * We note our dataset includes all non-combination FDA-approved drugs and is larger than other widely used human in-vivo toxicity datasets. Even our benchmarking subset is similar in size to the ClinTox dataset in Therapeutic Data Commons (a NeurIPS 2021 paper) which contains 1,484 drugs and a single binary Toxic/Safe rating per drug. The DILI dataset in TDC contains 475 drugs and LiverTox contains 1,507 drugs. Several peer-reviewed machine learning papers have trained or tested on smaller datasets of just FDA-approved drugs, including on DICTrank with __1318 drugs__ (PMID: 38300851), DILIrank with __1036 drugs__ (PMID: 32422053), DILIst with __1148 drugs__ (Narayanan et al. 2023, IEEE), and __923 drugs__ from DILIrank and SIDER4 (PMID: 33461600).
> * __Exclusion of biologics from our benchmarking:__ We agree that benchmarking performance across both small molecules and biologics would be preferable. Unfortunately, most models are not able to effectively handle both small molecules and larger molecules like antibodies. While we could train a separate biologics model, the small sample size (284 biologics in UniTox) would likely lead to poor performance. Excluding biologics from molecular classifiers is a common practice in the literature (the papers cited above all exclude biologics)
> * __Elaboration on Categories:__ We selected our eight toxicity categories in consultation with the clinician co-authors on the paper. The criteria was to include a broad range of organ systems where clinicians would most want standardized toxicity information
> * Finally, we agree with the reviewer that we can better elaborate on the limitations of our work. We added to our limitations:
>   * Despite adding additional model types to our benchmarks, these models are meant to predict from SMILES codes and a set of physicochemical features
>   * Our dataset is based on FDA-approved drugs and our benchmarking is restricted to small molecules to enable training a single model
>   * Our human validation consists of comparing to FDA expert-developed datasets where possible (cardiotoxicity, liver toxicity, renal toxicity) and clinician evaluation of 200 randomly selected samples for the other toxicities

---

> > ### Comment · Reviewer_ibpw · 2024-08-29
> > **Most concerns satisfactorily addressed**
> >
> > Thank you for your detailed response to my review. Several concerns have been satisfactorily addressed, including benchmarking with a broader range of models, incorporating parameters such as dosage, improving human evaluation reliability, and elaborating on the categories.
> >
> > While the size of the dataset and the exclusion of biologics remain challenges for model training, I understand the inherent bottlenecks in the data creation process, which are also seen in other works addressing similar issues.
> >
> > Based on the rebuttal, I have raised my rating and recommend the acceptance of the paper. I request the authors to incorporate the feedback in the camera-ready version.

---

> > > ### Author Response · Authors · 2024-08-30
> > > **Thank you!**
> > >
> > > We thank the reviewer again for their helpful feedback and suggestions that have improved our paper, and we are glad the reviewer found our additional information in the rebuttal to be helpful

---

> ### Author Rebuttal · Authors · 2024-08-26
>
> Dear Reviewer ibpw,
>
> Thank you very much for your feedback! We really appreciate it! We have carefully conducted additional experiments to address your comments. Please let us know if you have any further questions and we are happy to follow up. If we have addressed your questions, we would very much appreciate it if you would consider raising your score to reflect it. Thanks you!

---

### Official Review · Reviewer_scL6 · 2024-07-27

**Rating:** 7
**Confidence:** 3
**Correctness:** I think most of the claims in this ar…
**Clarity:** Yes.

**Review:**

This work addresses the current lack of standardized and systematic drug toxicity data organization in the field of drug discovery, aiding the AI community in better understanding drug toxicity. The authors utilize one of the most advanced models, GPT-4o, to summarize and rate FDA drug toxicity, and conduct a comprehensive evaluation of the rating quality from multiple dimensions, notably including clinical validation through wet lab experiments for missing parts, which is highly valuable and convincing. Overall, the motivation for constructing the dataset is clear, the organization process is standardized, and the usage instructions are clear. I believe this work is highly valuable to the AI for drug community. However, I have some concerns about the dataset evaluation, and some parts of the article are not clearly described, which I will detail in the following review.

**Strengths:**

- The article is logically clear and easy to read, with clearly stated problem and motivation.

- This article attempts to address an important research issue in AI for drug discovery—the systematic organization of multi-organ system toxicities in FDA drugs. Most current work on drug property prediction ignores the importance of data quality, and I appreciate that the authors have focused on this issue, providing a good research foundation for the healthy development of this field.

- The dataset construction process is sound and reasonable. Since the most critical step in constructing this dataset is the understanding and extraction of drug labels, it is very suitable to use a powerful LLM for this task. The data source is reliable, and the data cleaning and deduplication operations are reasonable. Additionally, the dataset is user-friendly, with clear usage instructions.

- The evaluation of dataset quality is comprehensive and reliable. The authors evaluate the quality of LLM outputs based on existing high-quality datasets. More importantly, they supplemented missing evaluation labels with additional clinical experiment validation, which all demonstrated high data annotation quality, further ensuring the reliability of this dataset in evaluating drug safety.

**Additional Feedback:**

I suggest the authors provide a more formal definition or statement of the **labels** and **ratings** mentioned in the paper, as the term "label" can mislead readers into thinking it refers to the final LLM-processed annotations, whereas in this paper, the ratings are the LLM  annotations.

**Documentation:**

Yes.

**Ethics:**

No ethical concerns.

**Limitations:**

Please refer to Opportunities For Improvement.

**Opportunities For Improvement:**

- The description of model output confidence in Section 2.2 is inaccurate. The authors suggest that the model's rating of drug toxicity can be seen as the model's confidence, which I believe is inaccurate or even incorrect. Uncertain outputs from the LLM cannot represent confidence, and conversely, seemingly certain outputs do not necessarily have high confidence. Simple prompt engineering can reduce the frequency of "less" outputs from the model, making it difficult to directly evaluate the confidence of the model’s answers from a single output. Therefore, I think the authors need to revise related statements or provide relevant statistical analysis to verify that this assumption holds true in such research scenario.

- Some descriptions of the dataset are unclear. Do the 2,418 FDA drugs in the UniTox dataset have ratings for all types of toxicity? If so, this would mean that all drug labels have descriptions of all toxicity types for the model to summarize and rate. I have concerns about the completeness of the label information because the prompt used by the authors is a hard demand. In an extreme case, there might be no related information about a particular toxicity in the label, yet it still provides a confident answer. Have the authors done any related statistics and checks to ensure this point?

- There are some doubts about the dataset evaluation.

1.  In Table 2, the authors only provide the number of drug samples for **Ternary w/o Less** but do not provide relevant category statistics for the referenced existing datasets. For example, in line 149, what proportion do the samples in **Ambiguous-DICT-Concern** and **Less-DICT-Concern** categories account for? If these proportions are large, the evaluation accuracy might have a significant bias because the ground truth itself is highly uncertain.

2. In the ablation experiments provided by the authors, I noticed an interesting phenomenon: the number of drugs in **Ternary w/o Less** significantly decreases when CoT is removed, and the accuracy also significantly drops. However, in the **Binary on GT No/Most** case, accuracy does not significantly decrease (though the number of drugs is not provided). Can the authors explain why the performance trends differ in these two prompt settings?

- NPV and PPV in Table 2 lack formal definitions in the text. If I understand correctly, they should be Negative Predictive Value and Positive Predictive Value, respectively, but the authors need to explain this instead of presenting them as abbreviations in the table, which is not conducive to reader understanding.

**Relation To Prior Work:**

Yes.

**Summary And Contributions:**

The paper introduces UniTox, a systematic drug-induced toxicity dataset curated using GPT-4o to extract toxicity information from FDA drug labels, covering eight toxicity types for 2,418 drugs. The dataset's reliability is validated by high clinician agreement and benchmarking against existing datasets. A Graph Neural Network trained on UniTox demonstrates its potential for enhancing toxicity prediction models, addressing a critical gap in comprehensive toxicity data. The authors discuss limitations and propose future enhancements, highlighting UniTox's significant contribution to drug safety evaluation.

---

> ### Author Rebuttal · Authors · 2024-08-16
>
> We thank the reviewer for these comments and suggestions. Below, please find the improvements we made in response, and a response to the reviewer’s questions:
>
> * __Reframing Model “Confidence”:__ We appreciate the suggestion and revised our framing, removing any reference to “confidence”: The ternary prompt allows the model to separate potential “borderline” cases of mild or very rare adverse reactions from more “clear cut” cases of either significant toxicity or no risk of toxicity. If performing well, the LLM will classify as “Less" toxic drugs where reasonable readers may disagree about whether a rare or mild drug reaction rises to the level of “Toxicity.” As a result, the model may have its worst accuracy (compared to clinician validations) on this predicted “Less” category, and better accuracy on the “No” and “Most” categories. So, for a novel toxicity without FDA-created validation data, using just the “No” and “Most” categories would minimize rating noise when training a new molecular classifier on UniTox ratings. We hope the reviewer agrees that this phrasing is more accurate and clear.
>
> * __Ratings for all drugs:__ The dataset indeed provides ratings for all 2,418 drugs for all toxicities. We note that the FDA-approved label is a comprehensive document (over 8,000 words long on average) that discusses all observed toxicity (e.g., adverse reactions in a clinical trial) as well as theoretical toxicity risks based on the mechanism of action (e.g., if other drugs in a similar drug class caused toxic reactions, even if this drug itself did not). Using cardiotoxicity as an example, if the FDA-approved label contains no mentions of cardiotoxic symptoms, that means that very few or no patients in any clinical trial of the drug experienced a cardiotoxic reaction attributable to the drug, and the drug’s mechanism of action has no association with cardiotoxicity. It also means that the FDA has not seen any evidence of cardiotoxic reactions in post-marketing surveillance of the drug since it was approved. Based on this level of comprehensiveness, the FDA authors of DICTrank and askFDALabel feel comfortable concluding that no mention of cardiotoxic symptoms or conditions in an FDA-approved drug label means a drug should be classified as having “No DICT Concern,” and we follow their lead. We thus feel confident applying this standard to other toxicities where any observed or theoretical toxicity during the FDA’s drug review would be mentioned in the FDA label.
>
> * Please note there is one exception where we deviate from this: because drugs are usually not tested on pregnant patients, we have the LLM separately clarify whether the drug was studied in these populations for our Infertility risk category and include this information in our Infertility data. This deviation was suggested by our clinician co-authors.
>
> * __Less and Ambiguous classifications in DICTrank:__ For DICTrank, 473 of 1,181 drugs (40%) were classified by the FDA as Less DICT Concern. However, these are not uncertain ratings, but instead correspond to an association with a specific set of milder cardiotoxic symptoms (e.g., tachycardia, palpitations). We follow the lead of other machine learning papers in binarizing “Less-DICT-Concern” drugs as positive cases (See PMID: 38300851). “Ambiguous” risk of Toxicity in DICTrank accounts for only 96 of 1,181 matches (8%). Our DICTrank results would improve if we excluded the “Ambiguous” drugs. Ternary accuracy would increase to 85% and our Ternary Without Less accuracy would increase to 94% (N = 701).
>
> * With regards to sample sizes in Table 2, please see in the attached PDF a revised Table 2 that clarifies N for all results.
>
> * __Chain-of-Thought:__ We thank the reviewer for noting this result. The LLM without Chain-of-Thought reads the full FDA drug label and provides a one word rating of predicted toxicity. In both the Ternary and Binary settings, the no-CoT model had a particular failure mode: drugs whose ground-truth classification was “Less” were predicted as “No” Cardiotoxicity. For example, the LLM would classify mild cardiotoxic reactions (e.g., hypertension) as “No”, whereas the FDA authors of DICTrank consider that to be “Less-DICT-Concern”. Because this specific failure is for ground-truth “Less” drugs, these are filtered out in the Binary GT No/Most column. So performance is little changed from our full pipeline. However, this failure is when the model predicts “No”, so these errors are __not__ filtered from the Ternary w/o Less column (where we only filter out the model’s prediction of Less). As a result, this failure mode significantly affects the Ternary w/o Less performance. We have modified the name of this column to Ternary on Predicted No/Most to clarify the prediction vs. ground-truth difference.
>
> * Separately, the No Chain-of-Thought ablation in the Ternary setting has a second challenge where it mistakes some severe cardiotoxic symptoms (e.g., myocardial infarction) to be Less cardiotoxic. This causes the decrease in N in the Ternary w/o Less column compared to the full model, as here we filter out predicted Less. Because the Binary on GT No/Most column is based on the FDA’s ground-truth ratings (rather than any predictions from the model), the N in this column is model-agnostic. The N for all experiments is provided in the attached revised Table 2.
>
> * We thank the reviewer for noting the lack of definitions for Positive Predictive Value and Negative Predictive Value. We have now moved these statistics and their definition to a new table in our appendix. We included NPV and PPV to demonstrate that the model’s errors were relatively balanced, that is, it was not simply always guessing Most Toxicity.
>
> * Finally, we have clarified the terms “label” as the FDA-approved drug label, and “rating” as the model’s prediction, and appreciate the reviewer’s suggestion on this

---

### Decision · Program_Chairs · 2024-09-26

**Decision:**

Accept (Spotlight)

**Comment:**

This paper presents a unified dataset UniTox of toxicity summaries and ratings of 2,418 FDA-approved drugs by using GPT-4o over FDA drug labels. This is an interesting attempt of using LLMs for dataset curation, which may be inspiring to other researchers and areas. Generally, all reviewers gave positive opinion on the work, and the authors had careful rebuttal and discussions with the reviewers.